# Molecular Mechanisms of Aminoglycoside-Induced Ototoxicity in Murine Auditory Cells: Implications for Otoprotective Drug Development

**DOI:** 10.3390/ijms26146720

**Published:** 2025-07-13

**Authors:** Cheng-Yu Hsieh, Jia-Ni Lin, Yi-Fan Chou, Chuan-Jen Hsu, Peir-Rong Chen, Yu-Hsuan Wen, Chen-Chi Wu, Chuan-Hung Sun

**Affiliations:** 1Department of Otolaryngology, Head and Neck Surgery, Taichung Tzu Chi Hospital, Buddhist Tzu Chi Medical Foundation, Taichung 427213, Taiwan; chengyu384650@gmail.com (C.-Y.H.); neonlin0939@gmail.com (J.-N.L.); yifanchou29@hotmail.com (Y.-F.C.); cjhsu@tzuchi.com.tw (C.-J.H.); 2School of Medicine, Tzu Chi University, Hualien 970374, Taiwan; peirrong_chen@tzuchi.com.tw (P.-R.C.); hatsuyuki2001@yahoo.com.tw (Y.-H.W.); 3Institute of Clinical Medicine, National Taiwan University College of Medicine, Taipei 100233, Taiwan; jimchenchiwu@gmail.com; 4Department of Otolaryngology, National Taiwan University Hospital, Taipei 100225, Taiwan; 5Department of Otolaryngology, Head and Neck Surgery, Hualien Tzu Chi Hospital, Buddhist Tzu Chi Medical Foundation, Hualien 970473, Taiwan; 6Department of Medical Research, National Taiwan University Hospital Hsin-Chu Branch, Hsinchu 300195, Taiwan

**Keywords:** aminoglycosides, ototoxicity, mouse auditory cell lines, transcriptomic analysis

## Abstract

Aminoglycoside antibiotics are critical in clinical use for treating severe infections, but they can occasionally cause irreversible sensorineural hearing loss. To establish a rational pathway for otoprotectant discovery, we provide an integrated, three-tier methodology—comprising cell-model selection, transcriptomic analysis, and a gentamicin–Texas Red (GTTR) uptake assay—to guide the development of otoprotective strategies. We first utilized two murine auditory cell lines—UB/OC-2 and HEI-OC1. We focused on TMC1 and OCT2 and further explored the underlying mechanisms of ototoxicity. UB/OC-2 exhibited a higher sensitivity to gentamicin, which correlated with elevated OCT2 expression confirmed via RT-PCR and Western blot. Transcriptomic analysis revealed upregulation of PI3K-Akt, calcium, and GPCR-related stress pathways in gentamicin-treated HEI-OC1 cells. Protein-level analysis further confirmed that gentamicin suppressed phosphorylated Akt while upregulating ER stress markers (GRP78, CHOP) and apoptotic proteins (cleaved caspase 3, PARP). Co-treatment with PI3K inhibitors (LY294002, wortmannin) further suppressed Akt phosphorylation, supporting the role of PI3K-Akt signaling in auditory cells. To visualize drug entry, we used GTTR to evaluate its applicability as a fluorescence-based uptake assay in these cell lines, which were previously employed mainly in cochlear explants. Sodium thiosulfate (STS) and *N*-acetylcysteine (NAC) significantly decreased GTTR uptake, suggesting a protective effect against gentamicin-induced hair cell damage. In conclusion, our findings showed a complex ototoxic cascade involving OCT2- and TMC1-mediated drug uptake, calcium imbalance, ER stress, and disruption of PI3K-Akt survival signaling. We believe that UB/OC-2 cells serve as a practical in vitro model for mechanistic investigations and screening of otoprotective compounds. Additionally, GTTR may be a simple, effective method for evaluating protective interventions in auditory cell lines. Overall, this study provides molecular-level insights into aminoglycoside-induced ototoxicity and introduces a platform for protective strategies.

## 1. Introduction

Hearing loss is the most common sensory deficit, affecting approximately 5% of the global population. According to reports by the World Health Organization (WHO), by 2050, around 2.5 billion people will experience varying degrees of hearing loss, and at least 700 million will require hearing rehabilitation [1]. Sensorineural hearing loss is the most common type of hearing loss, and it can result from genetic factors, exposure to ototoxic agents, noise, infections, chronic diseases, autoimmune disorders, trauma, or aging [2,3,4]. Among these, ototoxicity poses a public health and societal burden, because it impairs communication abilities and impact quality of life [5,6,7].

Aminoglycosides (e.g., gentamicin, tobramycin, amikacin, neomycin, kanamycin) are effective antibiotics widely used to treat severe gram-negative bacterial infections [8,9]. Despite their indispensable clinical value, aminoglycosides can occasionally cause irreversible inner-ear damage in susceptible patients [8]. Reported rates of permanent hearing impairment after aminoglycoside therapy range from 2% to 25%, depending on cumulative dose, therapy duration, concomitant conditions, and pre-disposing genetic variants [9,10]. Understanding and preventing this form of drug-induced ototoxicity, therefore, carries clear clinical value.

The House Ear Institute–Organ of Corti 1 (HEI-OC-1) cell line is an immortalized auditory cell line derived from the organ of Corti in transgenic mice that expresses key hair-cell markers (myosin 7a, prestin) and undergoes apoptotic cell death through caspase-3 activation in response to ototoxic agents [11]. Because of its ease of culture and robust response, HEI-OC1 has been widely used as a standard in vitro model for investigating ototoxic and otoprotective effects [11,12,13]. UB/OC-2 is an immortalized mouse organ-of-Corti cell line derived from embryonic day-13 mice bearing a temperature-sensitive SV40 large T antigen (auditory hair cell precursors immortalized from the mammalian inner ear). It expresses key cochlear epithelial markers—myosin 7a, α9-acetylcholine receptor, and Brn3.1. UB/OC-2 cells are commonly used to investigate calcium signaling, ion currents, cell differentiation, and pharmacologic protection mechanisms [14,15,16,17,18,19,20].

Previous studies demonstrate that ototoxic drugs gain access to cochlear hair cells chiefly through mechano-electrical-transduction (MET) channels at stereocilia tips, which are formed by the transmembrane channel-like protein-1 (TMC1) complex and are likewise permeable to cisplatin [21,22]. A secondary entry pathway involves the organic cation transporter-2 (OCT2) [23]. Because the widely used HEI-OC1 mouse auditory cell line lacks detectable OCT2 [24,25], its ability to model OCT2-mediated uptake is limited; by contrast, the conditionally immortalized UB/OC-2 line expresses both TMC1 and OCT2, potentially offering a more comprehensive platform for mechanistic studies.

To develop rational otoprotective strategies, we therefore adopted a step-wise workflow. First, we compared and validated gentamicin-induced ototoxicity using two conditionally immortalized mouse auditory cell lines, HEI-OC1 and UB/OC-2, with a particular focus on TMC1 and OCT2. Second, we performed transcriptomic analysis (RNA-seq) of gentamicin-exposed cells to identify the key signal pathway influenced by ototoxic damage and verified these pathways at the protein level [26,27]. Finally, we translated these mechanistic insights into a quantitative gentamicin-conjugated Texas Red (GTTR) fluorescence assay, which enabled real-time visualization of drug entry and facilitated rapid screening of candidate otoprotectants. In our study, we used sodium thiosulfate (STS), known to mitigate cisplatin ototoxicity [28], and the antioxidant *N*-acetylcysteine (NAC) [29,30]. This integrated approach—model comparison, molecular dissection, and otoprotective validation—establishes both a mechanistic foundation and screening platform for identifying novel otoprotective strategies.

## 2. Results

### 2.1. Cytotoxic Evaluation of Gentamicin

To investigate the cellular and molecular mechanisms involved in gentamicin-induced ototoxicity, we used immortalized mouse auditory cell lines (HEI-OC1 and UB/OC-2) to obtain comprehensive data. First, we tested the cytotoxic effect of gentamicin in HEI-OC1 and UB/OC-2 cell lines using an MTT assay. Cells were exposed to gentamicin at concentrations ranging from 1 to 2.5 mM after 24 h of treatment, and the estimated IC_50_ values for gentamicin were approximately 2 mM in HEI-OC1 cells and 1.25 mM in UB/OC-2 cells (Figure 1A,B).

### 2.2. Transcriptome Analysis in HEI-OC1 Cells

To link gene-level changes to functional assays, the transcriptomic analysis serves as a mechanistic bridge between basic biology and the development of pathway-specific otoprotective strategies. Following the identification of the cytotoxic effects of gentamicin in both cell lines, we proceeded to conduct RNA sequencing to map gentamicin-induced gene expression changes and identified the predominant signaling pathways involved. The RNA sequencing analysis was performed on HEI-OC1 cells subjected to gentamicin treatment to assess transcriptomic modifications. Total RNA was extracted and purified using the TURBO DNA-free Kit (Thermo Fisher Scientific, Waltham, MA, USA) and library preparation was carried out using the Universal Plus mRNA-Seq Library Preparation Kit (Tecan Group Ltd., Männedorf, Switzerland). Differentially expressed genes (DEGs) were identified and subsequently analyzed through KEGG and GO enrichment analyses.

The KEGG analysis of DEGs exhibiting a ≥2-fold change identified 295 enriched pathways (Figure 2A), with 197 pathways upregulated after gentamicin treatment. Among the top 20 enriched pathways illustrated in the Figure 2A, key pathways such as the PI3K-Akt signaling pathway, the neuroactive ligand receptor pathway, and the NOD-like receptor signaling pathway were prominently highlighted. These pathways are associated with essential cellular processes, including cell survival, autophagy, apoptosis, and neuronal proliferation. The significant enrichment of the PI3K-Akt pathway highlights its critical role in gentamicin-induced ototoxicity, as it regulates cell survival and stress responses during damage.

The results of the GO enrichment analysis in terms of DEGs (≥2-fold change) showed 715 enriched terms (Figure 2B), which were classified into three main categories: biological processes (BP), cellular components (CC), and molecular functions (MF). Notably, in the BP category, the most enriched terms included G protein-coupled receptor activity. In the CC category, terms such as plasma membrane, membrane, extracellular space, extracellular region, and cell junctions were significantly enriched, indicating the active involvement of membrane dynamics in the cellular response to gentamicin exposure. In the MF category, the most enriched terms included olfactory receptor activity. In addition, processes related to the extracellular region-related processes were prominently upregulated, suggesting that intercellular signaling and matrix interactions may play a role in the mechanisms of damage. These results also provide important molecular insights into the pathways that may contribute to ototoxicity.

### 2.3. Gene Expression and Protein Analysis on TMC1 and OCT2

Building on the transcriptomic findings observed in HEI-OC1 cells, we next examined whether two key genes involved in aminoglycoside uptake, specifically TMC1 and OCT2, exhibited similar regulatory patterns in both HEI-OC1 and UB/OC-2 cells. Western blot analysis revealed a notable increase in TMC1 protein levels following gentamicin treatment in both cell lines (Figure 3A). These results imply that the mechano-electrical transduction (MET) channel, mainly constituted by TMC1, may serve as a major entry point for these ototoxic agents in cochlear hair cells.

Interestingly, OCT2 protein expression showed a distinct pattern: it was significantly upregulated in UB/OC-2 cells but remained undetectable in HEI-OC1. This cell line-specific difference was further verified by PCR results, which confirmed the presence of OCT2 mRNA exclusively in UB/OC-2, but not in HEI-OC1 (Figure 3B). Taken together, these observations underscore a fundamental divergence between UB/OC-2 and HEI-OC1 in the uptake of gentamicin, particularly through the OCT2 transporter, and emphasize the value of UB/OC-2 as a reliable model for investigating ototoxic mechanisms.

### 2.4. Protein Level Validation in UB/OC-2

To further validate the transcriptomic findings from HEI-OC1 cells, we conducted a detailed analysis of protein levels in UB/OC-2, which are uniquely characterized by robust OCT2 expression. The results from the KEGG analysis indicated that the PI3K-Akt signaling pathway was closely associated with gentamicin-induced ototoxicity. To validate the transcriptional observations presented in Figure 2A, we performed protein expression analysis in UB/OC-2 cells using Western blot analysis. Consistent with the transcriptional findings, gentamicin-treated UB/OC-2 cells also exhibited increased phosphorylation of both PI3K and Akt (Figure 4A), indicating activation of the PI3K-Akt pathway. PI3K/Akt is an important intracellular signaling pathway cascade that is activated by various stimuli and plays a vital role in processes such as cell proliferation, cell cycle regulation, apoptosis, and other relevant pathophysiological processes [31].

Olfactory receptors, which belong to a family of G protein-coupled receptors (GPCRs), mediate cellular responses through the activation of cAMP signaling, leading to calcium influx [32,33]. This observation agrees with prior research that aminoglycosides disturb Ca^2+^ flux—partly by blocking connexin-26 hemichannels in cochlear cells [34,35]. Notably, the GO analysis in the HEI-OC1 revealed significant enrichment in G protein-coupled receptor signaling pathways and olfactory receptor activity, processes that are known to modulate intracellular calcium homeostasis. Calcium influx induces endoplasmic reticulum (ER) stress, resulting in the upregulation of GRP78 and CHOP, which ultimately triggers apoptosis [36,37]. To confirm this effect of gentamicin on ER stress, we assessed the expression of ER stress marker proteins, specifically GRP78 and CHOP using Western blot analysis. Both GRP78 and CHOP were significantly upregulated after gentamicin treatment, suggesting that gentamicin may have altered calcium homeostasis and exacerbated ER dysfunction (Figure 4B). 

Gentamicin-induced cell death could be mediated by the PI3K/Akt signaling pathway and ER calcium mobilization. Simultaneously, apoptotic markers (cleaved caspase 3 and cleaved PARP) were also elevated (Figure 4C), indicating the activation of cell death pathways under gentamicin challenge. Taken together, these protein-level findings suggest that gentamicin induces cell death, which subsequently activates PI3K-Akt signaling, impairing ER calcium handling, and promoting apoptosis in UB/OC-2 cells.

### 2.5. PI3K-Akt Pathway in Gentamicin-Induced Ototoxicity

To further validate the transcriptomic and protein expression data indicating the involvement of the PI3K-Akt signaling pathway in gentamicin-induced ototoxicity, we evaluated the phosphorylation status of Akt (p-Akt) in UB/OC-2 cells subjected to gentamicin treatment (1.25 mM) with or without the PI3K inhibitors LY294002 (200 nM) or wortmannin (500 nM). As shown in Figure 5, gentamicin alone markedly reduced Akt activation, as indicated by a reduction in the p-Akt to total Akt ratio (p-Akt/Akt) to 0.68 compared to the untreated control. Co-treatment with LY294002 or wortmannin further decreased this ratio to 0.45 and 0.39, respectively, suggesting an additive inhibitory effect on Akt phosphorylation. Importantly, administration of LY294002 or wortmannin alone resulted in minimal changes in the p-Akt/Akt ratio (1.01 and 0.74, respectively), suggesting that PI3K-Akt signaling is primarily activated under gentamicin-induced stress rather than in baseline conditions. These results support the role of PI3K-Akt signaling in modulating gentamicin-induced cytotoxicity and suggest that inhibition of this pathway may exacerbate damage to hair cells.

### 2.6. GTTR Uptake and Protective Effects of STS/NAC

In order to enhance our understanding of the molecular signaling pathways involved, we further investigated the intracellular uptake of gentamicin and assessed the protective effects of antioxidant compounds. GTTR, a fluorescently labeled gentamicin conjugate, was used to visually assess the permeation of gentamicin into cells. Previous research has indicated that STS offers protective benefits against ototoxicity induced by cisplatin or carboplatin [28,38]. Therefore, to further explore potential protective interventions and to confirm the dynamics of drug entry, we employed GTTR-based assays in the UB/OC-2 cell line. We examined the protective effects of different drugs on gentamicin-induced GTTR uptake in the mouse cochlear UB/OC-2 cell line (Figure 6). In the control group, which did not receive gentamicin, there was no observable GTTR uptake, as evidenced by the absence of red fluorescence. Conversely, in the gentamicin-only group, a significant increase in GTTR uptake was noted, indicated by prominent red fluorescence. These results suggest substantial gentamicin internalization within the UB/OC-2 cells, which may lead to cellular damage. The results underscore the ability of gentamicin to penetrate cochlear cells and induce ototoxic effects. Notably, cotreatment with otoprotective agents, including STS and NAC, resulted in a significant reduction in GTTR uptake compared to the gentamicin-only group. The observed decrease in red fluorescence implies that these agents may inhibit gentamicin internalization and thereby offer protection to UB/OC-2 cells from ototoxic damage. These findings support the potential role of STS and NAC as protective agents against gentamicin-induced ototoxicity.

## 3. Discussion

### 3.1. UB/OC-2 vs. HEI-OC1 as In Vitro Models

Both UB/OC-2 and HEI-OC1 are conditionally immortalized auditory cell lines derived from transgenic mice that express temperature-sensitive SV40 T antigen [11]. UB/OC-2 cells are derived from embryonic cochlear epithelium and retain progenitor-like characteristics, making them suitable for studying early hair cell development and molecular regulation in supporting cells. In contrast, HEI-OC1 cells, which are sourced from neonatal auditory tissue, have been extensively utilized in research focused on drug-, noise-, and age-related ototoxicity due to their partially differentiated phenotype [39,40].

In our study, transcriptomic analysis indicated that HEI-OC1 cells did not exhibit detectable levels of the organic cation transporter OCT2, which is a critical mediator of aminoglycoside uptake. Meanwhile, UB/OC-2 cells demonstrated expression of OCT2, thus seeming to provide a more accurate model for examining gentamicin transport mechanisms in cochlear tissues. Furthermore, both cell lines showed an upregulation of TMC1 expression following exposure to gentamicin, which aligns with its hypothesized role in facilitating aminoglycoside permeation through the MET channel.

Overall, these findings highlight the importance of integrating transcriptional profiling in HEI-OC1 with protein-level validation in UB/OC-2 to achieve a comprehensive understanding of ototoxic responses. Given the distinct expression patterns of OCT2 and TMC1, UB/OC-2 may be a more suitable model for exploring the uptake-dependent mechanisms of aminoglycoside-induced hair cell damage.

### 3.2. Key Insights from Transcriptomic and Protein Expression Analyses

Our transcriptomic analysis of HEI-OC1 cells revealed significant upregulation of genes associated with calcium channel activity and the PI3K-Akt signaling pathway, indicating a coordinated cellular response to aminoglycoside-induced stress. GO enrichment analysis further suggested the involvement of G protein-coupled receptor (GPCR) signaling, with certain olfactory receptor-related GPCRs potentially activating phospholipase C (PLC) and subsequently triggering IP_3_-mediated calcium release from the endoplasmic reticulum (ER). This signaling cascade appears to amplify ER stress under gentamicin-induced insult.

Calcium homeostasis is essential for auditory cell function, however, dysregulation—whether through the overactivation of calcium channels or the influx of gentamicin via MET channels—can result in excessive intracellular calcium (Ca^2+^) accumulation. Our transcriptomic analysis indicated an upregulation of genes related to calcium channels, suggesting a potential mechanism by which calcium influx may be intensified under gentamicin stress. Furthermore, gentamicin may also enter or bind to MET channels, disrupting normal ion flux and further increasing cytoplasmic Ca^2+^ [41,42,43]. This elevation may trigger a cytotoxic cascade characterized by ROS production, mitochondrial dysfunction, and activation of apoptotic signaling pathways. Given its role as a primary calcium reservoir, the endoplasmic reticulum (ER) is particularly vulnerable to calcium imbalance, often leading to unfolded protein responses and the upregulation of stress markers such as GRP78 and CHOP [44,45]. Taken together, these transcriptomic shifts, in conjunction with decreased Akt phosphorylation, suggest a complex ototoxic mechanism involving calcium overload, ER stress, and compromised cell survival signaling, ultimately leading to hair cell damage.

### 3.3. Further Implications of the PI3K-Akt Pathway in Gentamicin Ototoxicity

Based on the transcriptomic findings, we further investigated the role of the PI3K-Akt signaling pathway because of its central function in modulating stress responses and cell survival. In our study, gentamicin markedly reduced phosphorylated Akt levels, suggesting attenuation of survival signaling. This effect was further amplified when cells were co-treated with PI3K inhibitors, thereby implying that PI3K-Akt signaling functions as a protective mechanism during toxic stress conditions [46]. These pathways highlight potential targets for functional validation.

Recent studies indicates that the PI3K-Akt pathway interacts with multiple stress-related mechanisms; notably, inhibition of this pathway has been linked to enhanced ER dysfunction and increased expression of stress markers like GRP78 and CHOP, and it may intensify apoptotic responses [47]. When the PI3K-Akt pathway is suppressed and ROS are produced along with calcium imbalance, it can lead to serious damage inside the cell. These combined effects may make cochlear hair cells more vulnerable to gentamicin exposure [48].

Notably, recent studies have proposed activation of the PI3K-Akt pathway as a promising therapeutic strategy. Therefore, compounds such as (−) butaclamol have shown potential in decreasing aminoglycoside-induced cytotoxicity by enhancing prosurvival signaling pathways [49]. Although these results are still early-stage, they still support the rationale for targeting the PI3K-Akt pathway and anti-ROS agents to improve otoprotection strategies.

### 3.4. Implications of the GTTR Uptake Assay

We used GTTR fluorescence to visualize gentamicin uptake in UB/OC-2 cells. A strong intracellular signal indicated substantial drug entry. When pre-treated with sodium thiosulfate (5 µM) or *N*-acetylcysteine (2.5 mM), the fluorescence intensity reduced by ~40% (*p* < 0.05). This observation visually demonstrated that sodium thiosulfate and *N*-acetylcysteine reduced the entry of gentamicin into cells [50].

Although the protective effects of STS and NAC appear largely independent of PI3K-Akt signaling, they likely act as ROS scavengers. These results suggest a different therapeutic direction—one that aims to limit drug entry and strengthen cellular barriers, rather than downstream signaling cascades.

Furthermore, GTTR and related tracers, such as FM1-43, serve not only as tools to visualize drug accumulation, but also as practical screening tools for evaluating potential otoprotectants. When combined with assessments of PI3K-Akt activity and ER stress markers, these findings may support the development of comprehensive strategies for multifactorial otoprotection.

### 3.5. Species-Specific Genetic Variation

Missense mutations in TMC1, such as p.M418K and p.D572N in humans [51,52] and the “Beethoven” p.M412K mutation in mice, can increase MET-channel permeability and heighten sensitivity to aminoglycosides [53]. For OCT2, the common human A270S (rs316019) variant lowers transporter activity and alters cisplatin-related toxicity, whereas wild-type mouse Oct2 has higher drug affinity [23]. Because HEI-OC-1 and UB/OC-2 come from C57BL/6J mice with wild-type alleles, our models reflect baseline uptake rather than variant-specific behavior. Future work with CRISPR-engineered variants or patient-derived iPSC hair cells should clarify genotype-dependent transport and aid personalized otoprotection.

### 3.6. Future Directions

GTTR-based assays show promise as a high-throughput platform for evaluating the efficacy of MET channel blockers. By quantifying intracellular fluorescence, researchers can rapidly assess if candidate compounds affect gentamicin entry into auditory cells. In addition, transcriptomic analysis and molecular modeling can help uncover key gene networks and structural interactions involved in ototoxicity. These can be extended to test potential pathway-specific inhibitors to mitigate inner-ear damage.

Looking ahead, future work could focus on TMC1 and OCT2 to clarify their roles in drug uptake. Genetic manipulation strategies, such as CRISPR/Cas9 or shRNA, may also help identify downstream signaling cascades and find new therapeutic targets. We also agree that incorporating rat systems would strengthen translational relevance; therefore, integration with ex vivo rat cochlear explants and in vivo rodent models, followed by functional hearing assessments (e.g., ABR, DPOAE), will be critical for confirming the efficacy of emerging otoprotective strategies.

### 3.7. Practical Implications

Cell model selection:

UB/OC-2 cells, which express both TMC1 and OCT2, are recommended for studies that target transporter-mediated entry routes or screen OCT2 blockers. HEI-OC-1 cells, lacking OCT2, remain useful for work focused on MET channel pharmacology.

Assay window:

A 24 h exposure to 1.0–2.0 mM gentamicin produces a clear, graded loss of viability in UB/OC-2 cells and generates a robust gentamicin–Texas Red (GTTR) fluorescence signal, providing a practical range for otoprotective screening.

Mechanistic insight:

Based on the Western blots and inhibitor tests, our RNA-seq results identify PI3K–Akt activation as a predominant injury pathway, offering potential targets for pathway-specific drug discovery.

Otoprotectant evaluation:

Using the GTTR assay as a visual readout, both STS and NAC showed qualitative reductions in intracellular gentamicin fluorescence and preserved cell morphology, suggesting an effective method for otoprotectant screening.

Overall, by combining optimized cell models, transcriptomic profiling, and a GTTR-based uptake assay, this study provides valuable mechanistic insight and a practical screening platform for identifying novel otoprotective strategies.

## 4. Materials and Methods

Methodological Overview

This study followed a five-step workflow that links how gentamicin enters hair cells to the downstream pathways that induce ototoxicity, and builds a platform for screening protective strategies.

Step 1—Cell-line validation.

We compared two conditionally immortalized mouse auditory cell lines—HEI-OC1 and UB/OC-2—to determine which one more accurately models clinical gentamicin uptake.

Step 2—Transcriptomic analysis.

Gentamicin-treated versus control cells were subjected to RNA-seq. Differential expression was analyzed to identify key pathways linked to gentamicin susceptibility.

Step 3—Pathway validation and functional inhibition.

Key signaling pathways highlighted by RNA-seq were validated at the protein level (Western blot). Selected pathways (PI3K-Akt) were confirmed by Western blot and specific inhibitors (LY294002).

Step 4—GTTR-based screening.

Mechanistic insights were translated into a fluorescence gentamicin-Texas Red (GTTR) uptake assay, enabling real-time visualization of drug entry and rapid screening of candidate protectants. As proof of concept, we screened two compounds—sodium thiosulfate (STS) and the antioxidant *N*-acetylcysteine (NAC).

### 4.1. Cell Culture and Drug Treatment

The HEI-OC1 mouse cochlear hair cell line (kindly provided by Dr. Chen-Chi Wu) [54,55] was grown in high-glucose Dulbecco’s Eagle’s medium supplemented with GlutaMAX (Thermo Fisher Scientific, Waltham, MA, USA), 10% fetal bovine serum (FBS, Thermo Fisher Scientific, Waltham, MA, USA), and 50 U/mL γ-interferon (R&D systems, Minneapolis, MN, USA) at 33 °C in a humidified atmosphere of 5% CO_2_ [30,54,55]. The other mouse cochlear cell line, UB/OC-2, was purchased from Ximbio (London, UK) and cultured in Minimum essential media supplemented with GlutaMAX (Thermo Fisher Scientific, Waltham, MA, USA), 10% FBS, and 50 U/mL γ-interferon at 33 °C in a humidified atmosphere of 5% CO_2_ [18,56].

Cells were seeded into 6 cm dishes at a density of 1 × 10^6^ cells/well with 3 mL media and allowed to adhere overnight. The following day, cells were treated with varying concentrations of gentamicin (Standard Chemical & Pharmaceutical Co., Ltd., Tainan City, Taiwan) for 24 h for downstream analyses. Prior to gentamicin treatment, PI3K inhibitors LY294002 and wortmannin (MedChemExpress Ltd., Princeton, NJ, USA) were added to the culture medium.

### 4.2. Cell Viability Assay

The MTT (3-(4,5-dimethylthiazol-2-yl)-2,5-diphenyltetrazolium bromide, MedChemExpress Ltd., Princeton, NJ, USA) method was used to measure cell viability. Cells were seeded into 24-well plates at a density of 5 × 10^4^ cells/well and treated with various concentrations of gentamicin for 24 h. Cells were incubated in culture medium with MTT labeling solution at a final concentration of 0.2 mg/mL for 2 h at 33 °C. Then, formazan crystals were dissolved in dimethyl sulfoxide and the absorbance was detected at 570 nm using a microplate reader (Infinite 200 PRO Series Multimode Reader, Tecan Group Ltd., Männedorf, Switzerland). Cell viability of the control group was 100%.

### 4.3. RNA-Seq and Transcriptome Analysis

Immediately after gentamicin exposure, cells were placed in Trizol (Thermo Fisher Scientific, Waltham, MA, USA) for RNA isolation and then library construction using the Universal Plus mRNA-Seq Library Preparation Kit for Illumina sequencing (2 × 151 bp; NovaSeq 6000 sequencer, San Diego, CA, USA). Approximately 6 G bp of data were generated for each.

Quantification of raw reads was processed using CLC Genomics Workbench 10 software. The Illumina raw data were trimmed to remove adapters, low quality sequences (Q20) and ambiguous bases. The trimmed reads were used to perform de novo assembly using SPAdes (version 3.15.3) [57], and eukaryotic contigs were also removed. The filtered contigs were applied to perform rRNA prediction using RNAmmer (v1.2) [58], tRNA prediction using tRNAscan-SE (version 1.3.1) [59], and open reading frame (ORF) prediction using TransDecoder and Glimmer (version 3.02) [60]. The ORFs were annotated using the NCBI and COGs (Clusters of Orthologous Groups) database [61]. The ORFs were also annotated with gene ontology (GO) using FastAnnotation [62], pathways using KEGG Automatic Annotation Server (KAAS) [41], and antibiotic genes using CARD database [42,43].

The trimmed reads were mapped to ORFs using CLC Genomics Workbench. The mapping parameters were as follows: mismatch cost 2, insertion cost 3, deletion cost 3, length fraction of 0.5, and similarity fraction of 0.8. The expression values were calculated in Fragments Per Kilobases per Million (FPKM). The differential gene expression between two or more conditions was based on the fold change of the FPKM value. The genes with 2-fold change were further analyzed. The Kyoto Encyclopedia of Genes and Genomes (KEGG) database was used for pathway enrichment analysis, and the pathway map was plotted by pathview package [44] in R version 4.3.3. Gene ontology (GO) enrichment was analyzed by GO-TermFinder version 0.86 [45].

### 4.4. Western Blot Analysis

Total protein was extracted using ice-cold RIPA lysis buffer (Thermo Fisher Scientific, Waltham, MA, USA) containing protease inhibitor (Millipore Sigma, Oakville, ON, Canada) and phosphatase inhibitor (Sigma-Aldrich, St. Louis, MO, USA). Protein concentrations were determined using the BCA protein assay (VWR International, Radnor, PA, USA). Approximately 50 μg/sample was fractionated by SDS-PAGE (sodium dodecyl sulfate-polyacrylamide gel electrophoresis; VWR International, Radnor, PA, USA), and the separated proteins were transferred to polyvinylidene difluoride membranes (Merck, Darmstadt, Germany). The membranes were incubated in a blocking buffer containing 3% (*w/v*) bovine serum albumin in Tris-buffered saline (VWR International, Radnor, PA, USA) for 1 h at room temperature, followed by incubation with primary antibodies diluted 1:1000 at 4 °C overnight. The next day, the blots were incubated with secondary antibodies in TBS containing 0.1% Tween (TBST; VWR International, Radnor, PA, USA) for 1 h at room temperature. Primary antibodies against PI3K, phospho-PI3K (p-PI3K), Akt, phospho-Akt (p-Akt), caspase 3, cleaved Caspase 3 (c-Caspase 3), PARP, cleaved PARP (c-PARP), and β-actin were purchased from Cell Signaling Technology (Danvers, MA, USA). Primary antibodies against GRP78, CHOP, TMC1, and OCT2 were purchased from Abcam. HRP-conjugated secondary antibodies were purchased from PerkinElmer, Inc. (Shelton, CT, USA). Target proteins were visualized using enhanced chemiluminescence (Bio Rad Laboratories, Inc., Hercules, CA, USA) and imaged using a KETA C Chemi Imaging System (Wealtec Corporation, New Taipei City, Taiwan). The Image J software (version 1.52a, NIH, Bethesda, MD, USA) was used to analyze the expression of TMC1, OCT2, PI3K, p-PI3K, Akt, p-Akt, GRP78, CHOP, Caspase 3, c-caspase 3, PARP, and c-PARP, which were all normalized to β-actin.

### 4.5. RNA Isolation and Reverse Transcription-PCR (RT-PCR)

Total RNA was isolated using the RNeasy Mini Kit (Qiagen, Hilden, Germany) and QIAshredder (Qiagen, Hilden, Germany) according to the manufacturer’s recommended protocol. Reverse transcription-PCR (RT-PCR) of RNA samples was performed using Superscript III (Thermo Fisher Scientific, Waltham, MA, USA) and oligo-dT20 primers (Thermo Fisher Scientific, Waltham, MA, USA). PCR conditions were 94 °C for 1 min, followed by 35 cycles of denaturation at 95 °C for 30 s, annealing of specific primers at 58 °C for 30 s, and extension at 72 °C for 30 s, followed by a final extension at 72 °C for 10 min, and storage at 4 °C. Primers were designed as follows: OCT2, forward, TCTTGATGTACAATTGGTTCACG, and reverse, AACCACAGCAAATACGACCAG. These primers were designed to amplify a 461 bp fragment. The resulting PCR products were separated by electrophoresis on 2% agarose gels prestained with ethidium bromide (EtBr, Thermo Fisher Scientific, Waltham, MA, USA) and visualized under UV light illumination. Images were captured using a KETA M Series Imaging System (Wealtec Corporation, New Taipei City, Taiwan).

### 4.6. Gentamicin-Conjugated Texas Red (GTTR) Uptake

After seeding on chamber slides, cells were pretreated with various drugs for 2 h and then fed with gentamicin-conjugated Texas red (GTTR, AAT Bioquest, Pleasanton, CA, USA) in culture medium for 24 h at 33 °C. After incubation with GTTR for the desired times, cells were fixed with ice methanol (Thermo Fisher Scientific, Waltham, MA, USA) for 15 min at −20 °C. After washing with phosphate-buffered saline, fluorescence microscopy images were captured using an Olympus BX41 fluorescence microscope (Olympus Corporation, Tokyo, Japan).

### 4.7. Statistical Analysis

Statistical analysis was performed using SPSS version 22.0 software (IBM Corporation, Armonk, NY, USA), and data were presented as the mean ± standard deviation (SD) of at least three independent experiments. Data were tested for normality using the Shapiro–Wilk test. Differences were determined using one-way analysis of variance (ANOVA) followed by the Kruskal–Wallis test and Dunn’s test for multiple comparisons. *p* < 0.05 was considered a statistically significant difference.

## 5. Conclusions

In conclusion, this research clarifies the functions of TMC1 and OCT2 in ototoxicity within murine auditory cell lines and identifies UB/OC-2 as a more appropriate model for OCT2-dependent investigations. Transcriptomic and protein-level analyses suggest that gentamicin induces a multifactorial ototoxic response involving calcium dysregulation, ER stress, and impaired PI3K-Akt survival signaling. Additionally, GTTR uptake assays validated its utility as a simple, effective method for visualizing gentamicin entry. Preliminary screening with STS and NAC further demonstrates the platform’s capacity to evaluate candidate otoprotectants. Overall, by combining optimized cell models, transcriptomic profiling, and a GTTR-based uptake assay, these findings provide mechanistic insights into aminoglycoside-induced ototoxicity and support a platform for developing otoprotective strategies.

## Figures and Tables

**Figure 1 ijms-26-06720-f001:**
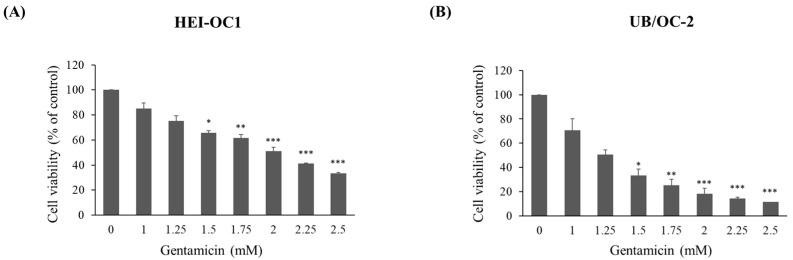
Effects of gentamicin on cell viability in HEI-OC1 and UB/OC-2 cells. (**A**) HEI-OC1 and; (**B**) UB/OC-2 cells (5 × 10^4^/well) were treated with 1, 1.25, 1.5, 1.75, 2, 2.25, and 2.5 mM of gentamicin for 24 h, and cell viability was measured using the MTT method. The results are expressed as percentages of the vehicle-treated control as 100%. Quantitative data are expressed as mean ± SD from three independent experiments in triplicates. * *p* < 0.05, ** *p* < 0.01, and *** *p* < 0.001 vs. the vehicle-treated control group.

**Figure 2 ijms-26-06720-f002:**
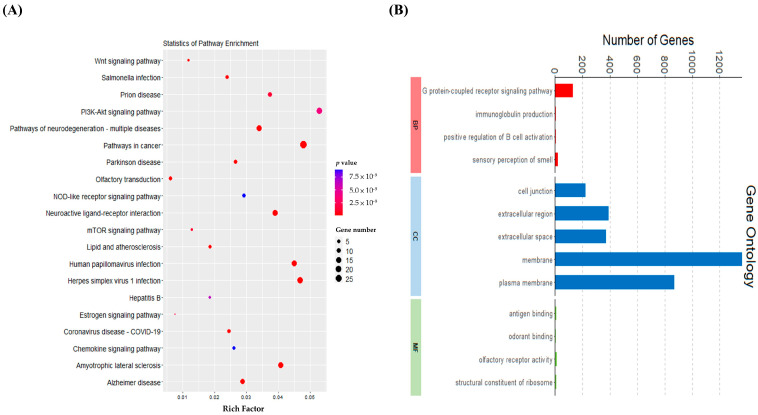
Enrichment analysis in HEI-OC1 cells. HEI-OC1 cells were treated with 2 mM gentamicin for 24 h and compared to untreated control cells. (**A**) KEGG pathway analysis: functional analysis of significantly upregulated genes. The vertical axis shows the annotated functions of the target genes, while the horizontal axis shows the enrichment score (gene ratio) and gene number of each cluster. Only the top 20 significantly enriched clusters are included; (**B**) GO terms based on biological processes (BP), cellular components (CC), and molecular functions (MF) for genes that change ≥2-fold).

**Figure 3 ijms-26-06720-f003:**
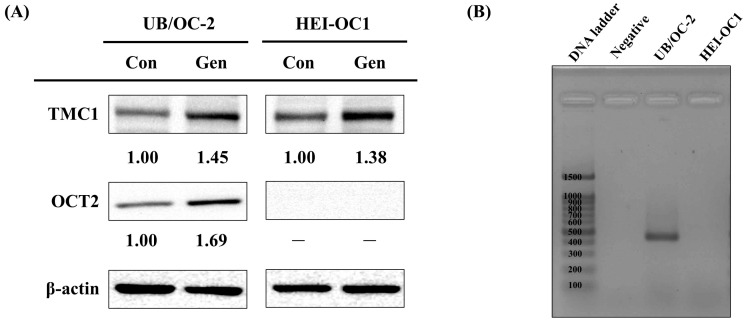
Protein and mRNA expression analysis of TMC1 and OCT2 after gentamicin treatment in UB/OC-2 and HEI-OC1 cells. Cells were treated with gentamicin at 1.25 mM in UB/OC-2 cells and 2 mM in HEI-OC1 cells for 24 h. (**A**) Western blot analysis showing TMC1 and OCT2 protein level increases after treatment with gentamicin in UB/OC-2 cells and HEI-OC1. β-actin serves as a loading control for all panels. The numerical values shown above each blot panel indicate the relative fold change in protein expression compared to the control group. Band intensities were quantified using ImageJ software version 1.52a; (**B**) OCT2 mRNA expression UB/OC-2 and HEI-OC1 cells by RT-PCR. Lane 1: DNA ladder (100–1500 bp); Lane 2 is negative control; lane 3 and lane 4 are PCR product from UB/OC-2 and HEI-OC1 cells, respectively. Bands of 461 bp are specific to UB/OC-2 cell line.

**Figure 4 ijms-26-06720-f004:**
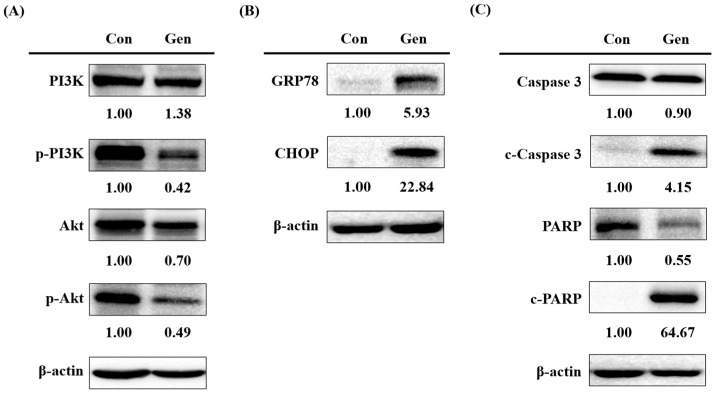
Effects of gentamicin in UB/OC-2 cells. (**A**) Western blot analysis of PI3K, phosphorylated PI3K (p-PI3K), Akt, and phosphorylated Akt (p-Akt) in control (Con) and gentamicin-treated (Gen) groups; (**B**) Expression levels of GRP78 and CHOP; (**C**) Western blot analysis was used to determine the expression levels of apoptosis-related proteins caspase 3, cleaved caspase 3 (c-caspase 3), PARP, and cleaved PARP (c-PARP). The numerical values shown above each blot panel indicate the relative fold change in protein expression compared to the control group. Band intensities were quantified using ImageJ software version 1.52a. β-actin serves as a loading control for all panels.

**Figure 5 ijms-26-06720-f005:**
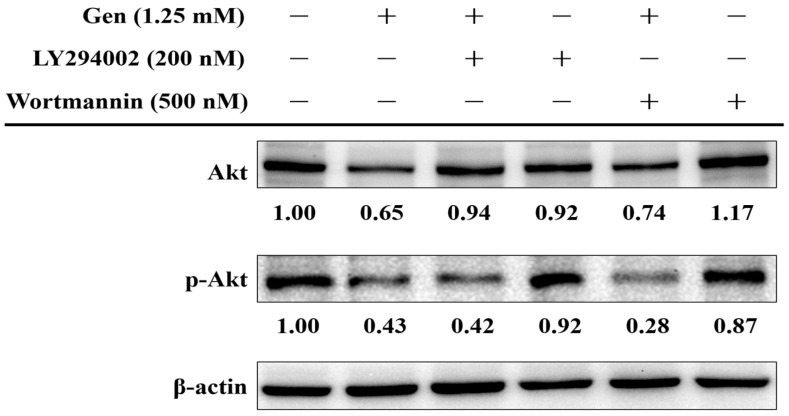
Effects of gentamicin and PI3K inhibitors on Akt phosphorylation in UB/OC-2 cells. UB/OC-2 cells were treated with gentamicin (1.25 mM), LY294002 (200 nM), or wortmannin (500 nM), alone or in combination, for 24 h. Western blot analysis was performed to detect total Akt and phosphorylated Akt (p-Akt). β-actin serves as a loading control for all panels. The numerical values shown above each blot panel indicate the relative fold change in protein expression compared to the control group. Band intensities were quantified using ImageJ software version 1.52a. Akt activation was quantified by calculating the p-Akt/Akt ratio, normalized to the control group (set as 1.00).

**Figure 6 ijms-26-06720-f006:**
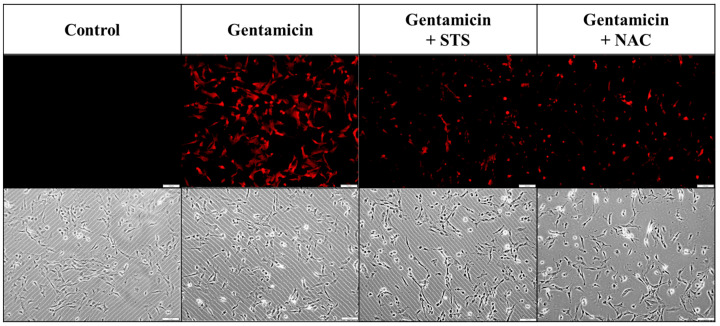
Gentamicin-induced GTTR uptake and protective effects of STS and NAC in UB/OC-2 cells. Cells were pretreated with 5 μM STS or 2.5 mM NAC for 2 h and then cotreated with 1.5 μg/mL GTTR (red) for 24 h. Scale bar = 100 µm.

## Data Availability

Data are contained within the article.

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
