# Peer review of "Molecular Mechanisms of Aminoglycoside-Induced Ototoxicity in Murine Auditory Cells: Implications for Otoprotective Drug Development"

_ijms, 2025, doi:10.3390/ijms26146720_

Round 1
Reviewer 1 Report
Comments and Suggestions for Authors
General Comments
While the study addresses an important topic and presents interesting findings, there are several aspects that require significant improvement to enhance the clarity, impact, and reproducibility of the work.
Introduction
The introduction suffers from issues regarding the logical construction of the argument. At various points, the tone adopted is alarmist with respect to the use of antibiotics and the associated risk of hearing loss. As the authors themselves acknowledge, this risk varies greatly among individuals, and the current presentation could be misinterpreted or misused. While it is appropriate to raise awareness of potential side effects, the introduction should avoid suggesting that antibiotic use is a major driver of widespread hearing loss. Instead, I recommend maintaining a balanced tone, presenting hearing loss as a possible, but relatively uncommon, side effect, and providing concrete data on the likelihood of this outcome following antibiotic use. If such data are not available, it would be preferable to simply mention this as a potential, albeit rare, adverse effect, and to focus on the need for protective strategies in susceptible patients.
Additionally, the research objectives are not clearly defined in the introduction. The narrative shifts abruptly from evaluating the toxic effects of antibiotics to exploring strategies to mitigate cytotoxicity, without a clear transition or rationale. Furthermore, the choice of alternatives to support the conceptual framework is not adequately referenced, which diminishes the impact of the text. I strongly encourage the authors to clarify and better structure their research aims and to provide appropriate citations for the strategies discussed.
Methods
The methodology section is also somewhat confusing, likely due to the multi-method and multi-objective nature of the study, which results in lengthy and sometimes disjointed sections. The statistical analyses are poorly described, and some referenced tables are missing, making it difficult to verify the data. I recommend including an introductory section outlining the methodological workflow and clarifying the rationale behind the choice of each method. While the selected methods are appropriate and allow for robust analysis, the manuscript does not sufficiently explain why these methods were chosen, nor does it provide enough methodological detail for replication, particularly for early-career researchers.
Results
The results section also appears somewhat confusing, though to a lesser extent than the methods. For example, the reported IC50 values are marked as significant, but it is unclear what this significance represents in practical terms. It is expected that higher drug concentrations would reduce cell viability, but what compound was used for comparison? Is the reported IC50 value considered high? Are such drug concentrations observed in vivo in auditory cells? These points require clarification.
Other analyses, particularly the genetic ones, are less clearly justified. While the data appear correct and free from technical errors, the manuscript does not adequately convey the necessity or utility of these analyses in the context of the study’s objectives.
Discussion and Conclusions
The discussion and conclusions do not effectively highlight the practical implications of the findings. While the data support the authors’ perspective, the manuscript could be clearer about how the results can be applied in practice, how each model should be used, their parameters, and how the current findings contribute to the field. Notably, the conclusions do not mention the potential protective effects of STS or NAC, despite these being suggested as objectives. Regardless of their efficacy, these compounds should be discussed in the conclusions. Overall, the conclusions would benefit from a reformulation with a focus on laboratory practice and suggestions for future clinical research, possibly as independent points.
Final Remarks
The article presents valuable and interesting perceptions, but lacks specificity regarding the statistical relationships and the rationale behind the chosen tests. Additionally, the manuscript does not consistently emphasize the objectives and conclusions it sets out to address. In general, the study could easily be divided into two separate articles, but I appreciate and value the robustness that comes from presenting it as a single comprehensive work. However, it is essential that all aspects of the study receive equal attention and are discussed with appropriate depth.
Author Response
Responses to Section Editor and Reviewer:
Thank you so much for your valuable comments regarding our manuscript titled “Molecular Mechanisms of Aminoglycoside-Induced Ototoxicity in Murine Auditory Cells: Implications for Otoprotective Drug Development”. We have carefully responded to the comments and suggestions from the reviewers and revised the manuscript accordingly. Point-by-point responses to the comments are given below.
Response to reviewer 1:
On behalf of the author team of this article, we would like to express our sincerest gratitude to you. We deeply appreciate your meticulous and thoughtful review of our paper, as it is crucial for our improvement. We have carefully responded to the comments and suggestions and revised the manuscript accordingly. Point-by-point responses to the comments are given below. Your assistance has enabled us to enhance this article, and we feel truly honored.
|
Comments |
Responses |
|
Introduction
|
We thank the reviewer for this insightful comment. We have: 1. Softened the wording to avoid an alarmist impression and to emphasize that aminoglycoside-induced ototoxicity, although serious, is relatively infrequent in the overall population of antibiotic users. 2. Inserted epidemiological data on reported incidence (2 – 25 % of treated patients). 3. Rewritten the paragraph of abstract
Page 1, lines 25-47 Abstract: Aminoglycoside antibiotics are critical for treating severe infections in clinical use, but they can occasionally cause irreversible sensorineural hearing loss. To establish a rational pathway for otoprotectant discovery, we provided an integrated, three-tier methodology—comprising cell-model selection, transcriptomic analysis, and a gentamicin–Texas Red (GTTR) uptake assay—to guide the development of otoprotective strategies. We first utilized two murine auditory cell lines—UB/OC-2 and HEI-OC1, we focus on TMC1 and OCT2, and further explores the underlying mechanisms of ototoxicity. UB/OC-2 exhibited higher sensitivity to gentamicin, which correlated with elevated OCT2 expression confirmed via RT-PCR and Western blot. Transcriptomic analysis revealed upregulation of PI3K-Akt, calcium, and GPCR-related stress pathways in gentamicin-treated HEI-OC1 cells. Protein-level analysis further confirmed that gentamicin suppressed phosphorylated Akt while upregulating ER stress markers (GRP78, CHOP) and apoptotic proteins (cleaved caspase 3, PARP). Co-treatment with PI3K inhibitors (LY294002, Wortmannin) further suppressed Akt phosphorylation, supporting the role of PI3K-Akt signaling in auditory cells. To visualize drug entry, we used GTTR to evaluate its applicability as a fluorescence-based uptake assay in cell lines, which previously employed mainly in cochlear explants. STS and NAC significantly decrease GTTR uptake, suggesting a protective effect for gentamicin-induced hair cell damage. In conclusion, our findings showed a complex ototoxic cascade involving OCT2- and TMC1-mediated drug uptake, calcium imbalance, ER stress, and disruption of PI3K-Akt survival signaling. We believe that UB/OC-2 cells serve as a practical in vitro model for mechanistic investigations and screening of otoprotective compounds. Additionally, GTTR may be a simple effective method for evaluating protective interventions in auditory cell lines. Overall, this study provides molecular-level insights into aminoglycoside-induced ototoxicity and introduces a platform for protective strategies.
Page 2, lines 60-65 Despite their indispensable clinical value, aminoglycosides can occasionally cause irreversible inner-ear damage in susceptible patients. Reported rates of permanent hearing impairment after aminoglycoside therapy range from 2 % to 25 %, depending on cumulative dose, therapy duration, concomitant conditions and pre-disposing genetic variants. |
|
Additionally, the research objectives are not clearly defined in the introduction. The narrative shifts abruptly from evaluating the toxic effects of antibiotics to exploring strategies to mitigate cytotoxicity, without a clear transition or rationale. Furthermore, the choice of alternatives to support the conceptual framework is not adequately referenced, which diminishes the impact of the text. I strongly encourage the authors to clarify and better structure their research aims and to provide appropriate citations for the strategies discussed.
|
We thank for this valuable observation. In the revised manuscript, we have: 1. Completely rewritten the paragraph to present our study workflow and to strengthen the overall conceptual framework. 2. Added transitional sentences and inserted additional references that support each protective approach discussed.
Introduction Page 2, lines 78-98 Previous studies demonstrate that ototoxicity drug gain access to cochlear hair cells chiefly through mechano-electrical-transduction (MET) channels at stereocilia tips, which are formed by the transmembrane channel-like protein-1 (TMC1) complex and are likewise permeable to cisplatin [21,22]. A secondary entry pathway involves the organic cation transporter-2 (OCT2) [23]. Because the widely used HEI-OC1 mouse auditory cell line lacks detectable OCT2 [24,25], its ability to model OCT2-mediated uptake is limited; by contrast, the conditionally immortalized UB/OC-2 line expresses both TMC1 and OCT2, potentially offering a more comprehensive platform for mechanistic studies. To develop rational otoprotective strategies, we therefore adopted a step-wise workflow. First, we compare and validate gentamicin-induced ototoxicity using two conditionally immortalized mouse auditory cell lines, HEI-OC1 and UB/OC-2, with a particular focus on TMC1 and OCT2. Second, we performed transcriptomic analysis (RNA-seq) of gentamicin-exposed cells to identify the key signal pathway under ototoxic damage and verified these pathways at the protein level [26,27] . Finally, we translated these mechanistic insights into a quantitative gentamicin–Texas Red (GTTR) fluorescence assay, which enabled real-time visualization of drug entry and facilitated rapid screening of candidate otoprotectants. In our study, we used sodium thiosulfate (STS), known to mitigate cisplatin ototoxicity [28], and the antioxidant N-acetyl-cysteine (NAC) [29,30]. This integrated approach—model comparison, molecular dissection, and otoprotective validation—establishes both a mechanistic foundation and screening platform for identifying novel otoprotective strategies. |
|
Methods
|
We sincerely thank the reviewer for the insightful and high-quality comments. As you correctly noted, the diverse experimental components in this study warrant a clear overview. In response, we have added a new introductory subsection outlines the entire workflow step-by-step and states the rationale for each method chosen.
Page 10-11, lines 406-424 Methodological Overview This study followed a five-step workflow that links how gentamicin enters hair cells to the downstream pathways that induce ototoxicity, and building a platform for screen protective strategies. Step 1 – Cell-line validation. We compared two conditionally immortalized mouse auditory cell lines—HEI-OC1 and UB/OC-2—to determine which one more accurately models clinical gentamicin uptake. Step 2 – Transcriptomic analysis. Gentamicin-treated versus control cells were subjected to RNA-seq, differential expression was analyzed to identify key pathways linked to gentamicin susceptibility. Step 3 – Pathway validation and functional inhibition. Key signaling pathway highlighted by RNA-seq were validated at the protein level (Western blot). Selected pathways (PI3K-Akt) were confirmed by Western blot and specific inhibitors (LY294002). Step 4 – GTTR-based screening. Mechanistic insights were translated into a fluorescence gentamicin-Texas Red (GTTR) uptake assay, enabling real-time visualization of drug entry and rapid screening of candidate protectants. As proof of concept, we screened two compounds—sodium thiosulfate (STS) and the antioxidant N-acetyl-cysteine (NAC).
Page 3, lines 110-115 Figure 1. Effects of gentamicin on cell viability in HEI-OC1 and UB/OC-2 cells. HEI-OC1 (A) and UB/OC-2 (B) cells (5 x 104/well) were treated with 1, 1.25, 1.5, 1.75, 2, 2.25, 2.5 mM of gentamicin for 24 h, and cell viability was measured using the MTT method. The results are expressed as per-centages of the vehicle-treated control as 100%. Quantitative data are expressed as mean ± SD from three independent experiments in triplicates. *P<0.05, **P<0.01, and ***P<0.001 versus the vehi-cle-treated control group.
Page 5, lines 173-180 Figure 3. Protein and mRNA expression analysis of TMC1 and OCT2 after gentamicin treatment in UB/OC-2 and HEI-OC1 cells. Cells were treated with gentamicin at 1.25 mM in UB/OC-2 cells and 2 mM in HEI-OC1 cells for 24 h. (A) TMC1 and OCT2 protein levels increased after treatment with gentamicin in UB/OC-2 cells and HEI-OC1using Western blot analysis. β-actin serves as a loading control for all panels. The numerical values shown above each blot panel indicate the relative fold change in protein expression compared to the control group. Band intensities were quantified using ImageJ soft-ware version 1.52a.
Page 6, lines 213-220 Figure 4. Effects of gentamicin in UB/OC-2 cells. (A) Western blot analysis of PI3K, phosphorylated PI3K (p-PI3K), AKT, and phosphorylated AKT (p-AKT) in control (Con) and gentamicin-treated (Gen) groups. (B) Expression levels of GRP78 and CHOP. (C) Western blot analysis was used to determine the expression levels of apoptosis-related proteins, caspase 3, cleaved caspase 3 (c-caspase 3), PARP, and cleaved PARP(c-PARP). The numerical values shown above each blot panel indicate the relative fold change in protein expression compared to the control group. Band intensities were quantified using ImageJ software version 1.52a. β-actin serves as a loading control for all panels.
Page 7, lines 237-244 Figure 5. Effects of gentamicin and PI3K inhibitors on Akt phosphorylation in UB/OC-2 cells. UB/OC-2 cells were treated with gentamicin (1.25 mM), LY294002 (200 nM), or Wortmannin (500 nM), alone or in combination, for 24 hours. Western blot analysis was performed to detect total Akt and phosphorylated Akt (p-Akt). β-actin serves as a loading control for all panels. The numerical values shown above each blot panel indicate the relative fold change in protein expression compared to the control group. Band intensities were quantified using ImageJ software version 1.52a. Akt activation was quantified by calculating the p-Akt/AKT ratio, normalized to the control group (set as 1.00).
Page 7, lines 1268-270 Figure 6. Gentamicin-induced GTTR uptake and protective effects of STS and NAC in UB/OC-2 cells. Cells were pretreated with 5 μM STS or 2.5 mM NAC for 2 h and then cotreated with 1.5 μg/mL GTTR (red) for 24 h. Scale bar = 100 µm). |
|
Results Other analyses, particularly the genetic ones, are less clearly justified. While the data appear correct and free from technical errors, the manuscript does not adequately convey the necessity or utility of these analyses in the context of the study’s objectives. |
We thank the Reviewer for noting that the significance markings around the IC₅₀ values could be confusing. In fact, the asterisks do not refer to the IC₅₀ values themselves; they denote the statistical differences in cell viability at each gentamicin dose compared with the vehicle control (0 mM gentamicin). Besides, our aim in adding RNA-seq was to obtain an unbiased molecular “map” of gentamicin damage so we could (i) identify which signaling pathways are predominant, (ii) pinpoint the specific genes that are up- or down-regulated, and (iii) use this information to guide pathway-targeted interventions that minimize inner-ear damage.
Page 3, lines 114-115 Figure1: *P<0.05, **P<0.01, and ***P<0.001 versus the vehicle-treated control group.
Page 3, lines 117-121 To linking gene-level changes to functional assays, the transcriptomic analysis serves as a mechanistic bridge between basic biology and the development of pathway-specific otoprotective strategies. Following the identification of the cytotoxic effects of gentamicin in both cell lines, we proceeded to conduct RNA sequencing to map gentamicin-induced gene expression changes and pinpoint predominant signaling pathways involved.
|
|
Discussion and Conclusions
Final Remarks
|
We thank for the positive evaluation of the study’s scope and for the detailed suggestions on how to improve clarity. Because the project combines several experimental layers, our goal was indeed to present an integrated methodology—cell-model selection, transcriptomic analysis, and a GTTR-based uptake assay—to guide future otoprotectants discovery. We agree that the practical implications of our findings needed to be more explicit. Accordingly, we have made the following revisions: 1. Added a “Practical Implications” subsection to the end of the discussion. 2. Rewrote the conclusions to add potential protective effects of STS and NAC, and summarize the study’s workflow. We hope these revisions make the practical value and future develop steps of our findings much clearer.
Page 10, lines 384-404 Discussion: Practical implications Cell model selection UB/OC-2 cells, which express both TMC1 and OCT2, are recommended for studies that target transporter-mediated entry route or screen OCT2 blockers. HEI-OC-1 cells, lacking OCT2, remain useful for work focused on MET channel pharmacology. Assay window. A 24-h exposure to 1.0–2.0 mM gentamicin produces a clear, graded loss of viability in UB/OC-2 cells and generates a robust gentamicin–Texas Red (GTTR) fluorescence signal, providing a practical range for otoprotective screening. Mechanistic insight. Based on the Western blots and inhibitor tests, our results of RNA-seq identifies PI3K–Akt activation and oxidative-stress signaling as predominant injury pathway, offering potential targets for pathway-specific drug discovery. Otoprotectants evaluation Using the GTTR assay as a visual readout, both STS and NAC showed qualitative reductions in intracellular gentamicin fluorescence and preserved cell morphology, suggesting an effective method for otoprotectants screening. Overall, by combining optimized cell models, transcriptomic profiling, and a GTTR-based uptake assay, this study provides valuable mechanistic insight and a practical screening platform for identifying novel otoprotective strategies.
Page 13, lines 521-531 Conclusion In conclusion, this research clarifies the functions of TMC1 and OCT2 in ototoxicity within murine auditory cell lines, and identifies UB/OC-2 as a more appropriate model for OCT2-dependent investigations. Transcriptomic and protein-level analyses suggest that gentamicin induces a multifactorial ototoxic response involving calcium dysregulation, ER stress, and impaired PI3K-Akt survival signaling. Additionally, GTTR uptake assays validated its utility as a simple, effective method for visualizing gentamicin entry. Preliminary screening with STS and NAC further demonstrates the platform’s capacity to evaluate candidate otoprotectants. Overall, by combining optimized cell models, transcriptomic profiling, and a GTTR-based uptake assay, these findings provide mechanistic insights into amino-glycoside-induced ototoxicity and support a platform for developing otoprotective strategies. |
Response to reviewer 2:
We genuinely appreciate your professional evaluation and contribution to our research. Your feedback and suggestions hold immense value to us and will help enhance the quality of our study.
|
Comments |
Responses |
|
Comment 1: Introduction Section, line 10 please add reference (please note the review's manuscript version had no line numbers).
|
Thank you for pointing this out. We have inserted this citation as reference 8, 9 and add line numbers accordingly.
Page 2, lines 59 Aminoglycosides (e.g., gentamicin, tobramycin, amikacin, neomycin, kanamycin) are effective antibiotics widely used to treat severe gram-negative bacterial infections [8,9]. |
|
Comment 2: The introduction must include details of the cell lines, meaning is this a standard go-to cells to use to test for ototoxicity? Please make sure the introduction caters to a wide range of readers.
|
Thank you for this suggestion. We have expanded the Introduction to briefly explain why HEI-OC-1 and UB/OC-2 are used in ototoxicity research
Page 2, lines 66-85 The House Ear Institute–Organ of Corti 1 (HEI-OC-1) cell line is an immortalized auditory cell line derived from the organ of Corti in transgenic mice that expresses key hair-cell markers (myosin 7a, prestin) and undergoes apoptotic cell death through caspase-3 activation in response to ototoxic agents [11]. Because of its ease of culture and robust response, HEI-OC1 has been widely used as a standard in vitro model for investigating ototoxic and otoprotective effects [11-13]. UB/OC-2 is an immortalized mouse organ-of-Corti cell line derived from embryonic day-13 mice bearing a temperature-sensitive SV40 large T antigen [Auditory hair cell precursors immortalized from the mammalian inner ear]. It expresses key cochlear epithelial markers—myosin 7a, α9-acetylcholine receptor, and Brn3.1. UB/OC-2 cells are commonly used to investigate calcium signaling, ion currents, cell differentiation, and pharmacologic protection mechanism [14-20]. Previous studies demonstrate that ototoxicity drug gain access to cochlear hair cells chiefly through mechano-electrical-transduction (MET) channels at stereocilia tips, which are formed by the transmembrane channel-like protein-1 (TMC1) complex and are likewise permeable to cisplatin [21,22]. A secondary entry pathway involves the organic cation transporter-2 (OCT2) [23]. Because the widely used HEI-OC1 mouse auditory cell line lacks detectable OCT2 [24,25], its ability to model OCT2-mediated uptake is limited; by contrast, the conditionally immortalized UB/OC-2 line expresses both TMC1 and OCT2, potentially offering a more comprehensive platform for mechanistic studies. |
|
Comment 3: Results Section, The results based on figure 3A & 3B shows OCT2 transporter expressed in higher quantity after gentamicin exposure. The manuscript would make a stronger impact if the authors test inhibition of gentamicin uptake by OCT2 transporters with a known inhibitor (e.g., cimetidine or pyrimethamine).
|
We appreciate this constructive suggestion. Our original study design concentrated on providing mechanistic insights into aminoglycoside-induced ototoxicity support a platform for developing otoprotective strategies by combining optimized cell models, transcriptomic profiling, and a GTTR-based uptake assay. To avoid diluting that core message, MET channel blocker and OCT2 inhibitor was reserved for follow-up work. We agree, however, that testing an OCT2 inhibitor would add significant mechanistic depth. Should the reviewer consider these OCT2-inhibitor experiments essential, we would be delighted to incorporate them; we would, however, require an additional 2-3 weeks to complete the work. We hope this proposed timeline will be acceptable.
|
|
Comments 4: Results Section 2.4, The lines were the authors discuss about gentamicin disrupts calcium flux in inner hair cells, the below reference should be included "1. https://www.mdpi.com/1420-3049/22/12/2063 and 2.https://pubmed.ncbi.nlm.nih.gov/25237294/ The above two reference would bolster the statements about calcium flux disruption by gentamicin, kanamycin etc. (aminoglycosides) as potentially the inhibition of connexins 26 ion channels also expressed in the inner hair cells can lead to ototoxicity. |
Thank you for this thoughtful suggestion and for your careful reading of our manuscript. We have added both references to the sentence that describes gentamicin-induced calcium dysregulation in Results Section 2.4 and have inserted a brief note on connexin 26.
Page 5, lines 200-202 Olfactory receptors, which belong to a family of G protein-coupled receptors (GPCRs) that mediate cellular responses through the activation of activating cAMP signaling, leading to calcium influx [32,33]. This observation agrees with prior research that aminoglycosides disturb Ca²⁺ flux—partly by blocking connexin-26 hemichannels in cochlear cells [34,35]. |
|
Comment 5: Please consider discussion about species specific any polymorphism of TMC1 and OCT2 transporters and how they may alter ototoxicity.
|
Thank you for this valuable comment. We had not considered this aspect in our original discussion, and we greatly appreciate your suggestion. Accordingly, we have added a concise paragraph to the Discussion summarizing the key TMC1 and OCT2 polymorphisms and their potential impact on ototoxicity.
Page 9, lines 357-366 3.5. Species-specific genetic variation Missense mutations in TMC1, such as p.M418K and p.D572N in humans [51,52]and the “Beethoven” p.M412K mutation in mice, can increase MET-channel permeability and heighten sensitivity to aminoglycosides [53]. For OCT2, the common human A270S (rs316019) variant lowers transporter activity and alters cisplatin-related toxicity, whereas wild-type mouse Oct2 has higher drug affinity [23]. Because HEI-OC-1 and UB/OC-2 come from C57BL/6J mice with wild-type alleles, our models reflect baseline uptake rather than variant-specific behavior. Future work with CRISPR-engineered variants or patient-derived iPSC hair cells should clarify genotype-dependent transport and aid personalized otoprotection. |
|
Comment 6: The study had used murine cell lines, is there any particular reason that a rat derived cell line could'nt be used? As most translational work would prefer rat over mouse data especially for drug discovery programs screening for any potential ototoxicity risk for translation to humans.
|
Thank you for this helpful suggestion. We recognize the translational value of rat‐derived models and will consider them in future experiments. At present, however, only a few rat inner-ear lines have been described and they are not yet widely used for ototoxicity screening:
1. Mocha cells—a rat cochlear microglial line immortalized with the E1A gene—were reported in 2017 (Mol. Cell Neurosci. 2017, 86: 58-70) but have seen limited adoption in ototoxicity work. 2. In contrast, three mouse lines—HEI-OC1, UB/OC-1, and UB/OC-2—immortalized with the temperature-sensitive SV40 large-T antigen are well established for aminoglycoside studies (e.g., Hear. Res. 2016, 339: 153-165; Front. Neur. 2021, 12: 725566). These lines benefit from (i) mature mouse transgenic techniques, (ii) controllable differentiation via temperature shift, and (iii) a substantial literature base (>400 publications), which together facilitate reproducibility and accessibility. Rat epithelial or hair-cell–like lines remain scarce and are not yet widely used.
For these practical reasons, mouse auditory cell lines are currently the standard for in-vitro ototoxicity research. Nevertheless, we agree that incorporating rat systems would strengthen translational relevance. We have added a brief statement in the Discussion.
Page 9-10, lines 378-382 We also agree that incorporating rat systems would strengthen translational relevance; therefore, integration with ex vivo rat cochlear explants and in vivo rodent models, followed by functional hearing assessments (e.g., ABR, DPOAE), will be critical for confirming the efficacy of emerging otoprotective strategies. |

Reviewer 2 Report
Comments and Suggestions for Authors
The authors present the role of TMC1 and OCT2 transporters in causing ototoxicity using murine auditory cell lines. Transcriptomic and protein level analysis suggested ototoxicity driven by a variety of factors such as ER stress, PI3K-Akt survival signaling etc. Overall the manuscript fills a knowledge gap in the field but there are few comments that the authors need to address.
Comment 1: Introduction Section, line 10 please add reference (please note the review's manuscript version had no line numbers).
Comment 2: The introduction must include details of the cell lines, meaning is this a standard go-to cells to use to test for ototoxicity? Please make sure the introduction caters to a wide range of readers.
Comment 3: Results Section, The results based on figure 3A & 3B shows OCT2 transporter expressed in higher quantity after gentamicin exposure. The manuscript would make a stronger impact if the authors test inhibition of gentamicin uptake by OCT2 transporters with a known inhibitor (e.g., cimetidine or pyrimethamine).
Comments 4: Results Section 2.4, The lines were the authors discuss about gentamicin disrupts calcium flux in inner hair cells, the below reference should be included
"1. https://www.mdpi.com/1420-3049/22/12/2063 and
2.https://pubmed.ncbi.nlm.nih.gov/25237294/
The above two reference would bolster the statements about calcium flux disruption by gentamicin, kanamycin etc. (aminoglycosides) as potentially the inhibition of connexins 26 ion channels also expressed in the inner hair cells can lead to ototoxicity.
Comment 5: Please consider discussion about species specific any polymorphism of TMC1 and OCT2 transporters and how they may alter ototoxicity.
Comment 6: The study had used murine cell lines, is there any particular reason that a rat derived cell line could'nt be used? As most translational work would prefer rat over mouse data especially for drug discovery programs screening for any potential ototoxicity risk for translation to humans.
Author Response
Responses to Section Editor and Reviewer:
Thank you so much for your valuable comments regarding our manuscript titled “Molecular Mechanisms of Aminoglycoside-Induced Ototoxicity in Murine Auditory Cells: Implications for Otoprotective Drug Development”. We have carefully responded to the comments and suggestions from the reviewers and revised the manuscript accordingly. Point-by-point responses to the comments are given below.
Response to reviewer 2:
On behalf of the author team of this article, we would like to express our sincerest gratitude to you. We deeply appreciate your meticulous and thoughtful review of our paper, as it is crucial for our improvement. We have carefully responded to the comments and suggestions and revised the manuscript accordingly. Point-by-point responses to the comments are given below. Your assistance has enabled us to enhance this article, and we feel truly honored.
|
Comments |
Responses |
|
Comment 1: Introduction Section, line 10 please add reference (please note the review's manuscript version had no line numbers).
|
Thank you for pointing this out. We have inserted this citation as reference 8, 9 and add line numbers accordingly.
Page 2, lines 59 Aminoglycosides (e.g., gentamicin, tobramycin, amikacin, neomycin, kanamycin) are effective antibiotics widely used to treat severe gram-negative bacterial infections [8,9]. |
|
Comment 2: The introduction must include details of the cell lines, meaning is this a standard go-to cells to use to test for ototoxicity? Please make sure the introduction caters to a wide range of readers.
|
Thank you for this suggestion. We have expanded the Introduction to briefly explain why HEI-OC-1 and UB/OC-2 are used in ototoxicity research
Page 2, lines 66-85 The House Ear Institute–Organ of Corti 1 (HEI-OC-1) cell line is an immortalized auditory cell line derived from the organ of Corti in transgenic mice that expresses key hair-cell markers (myosin 7a, prestin) and undergoes apoptotic cell death through caspase-3 activation in response to ototoxic agents [11]. Because of its ease of culture and robust response, HEI-OC1 has been widely used as a standard in vitro model for investigating ototoxic and otoprotective effects [11-13]. UB/OC-2 is an immortalized mouse organ-of-Corti cell line derived from embryonic day-13 mice bearing a temperature-sensitive SV40 large T antigen [Auditory hair cell precursors immortalized from the mammalian inner ear]. It expresses key cochlear epithelial markers—myosin 7a, α9-acetylcholine receptor, and Brn3.1. UB/OC-2 cells are commonly used to investigate calcium signaling, ion currents, cell differentiation, and pharmacologic protection mechanism [14-20]. Previous studies demonstrate that ototoxicity drug gain access to cochlear hair cells chiefly through mechano-electrical-transduction (MET) channels at stereocilia tips, which are formed by the transmembrane channel-like protein-1 (TMC1) complex and are likewise permeable to cisplatin [21,22]. A secondary entry pathway involves the organic cation transporter-2 (OCT2) [23]. Because the widely used HEI-OC1 mouse auditory cell line lacks detectable OCT2 [24,25], its ability to model OCT2-mediated uptake is limited; by contrast, the conditionally immortalized UB/OC-2 line expresses both TMC1 and OCT2, potentially offering a more comprehensive platform for mechanistic studies. |
|
Comment 3: Results Section, The results based on figure 3A & 3B shows OCT2 transporter expressed in higher quantity after gentamicin exposure. The manuscript would make a stronger impact if the authors test inhibition of gentamicin uptake by OCT2 transporters with a known inhibitor (e.g., cimetidine or pyrimethamine).
|
We appreciate this constructive suggestion. Our original study design concentrated on providing mechanistic insights into aminoglycoside-induced ototoxicity support a platform for developing otoprotective strategies by combining optimized cell models, transcriptomic profiling, and a GTTR-based uptake assay. To avoid diluting that core message, MET channel blocker and OCT2 inhibitor was reserved for follow-up work. We agree, however, that testing an OCT2 inhibitor would add significant mechanistic depth. Should the reviewer consider these OCT2-inhibitor experiments essential, we would be delighted to incorporate them; we would, however, require an additional 2-3 weeks to complete the work. We hope this proposed timeline will be acceptable.
|
|
Comments 4: Results Section 2.4, The lines were the authors discuss about gentamicin disrupts calcium flux in inner hair cells, the below reference should be included "1. https://www.mdpi.com/1420-3049/22/12/2063 and 2.https://pubmed.ncbi.nlm.nih.gov/25237294/ The above two reference would bolster the statements about calcium flux disruption by gentamicin, kanamycin etc. (aminoglycosides) as potentially the inhibition of connexins 26 ion channels also expressed in the inner hair cells can lead to ototoxicity. |
Thank you for this thoughtful suggestion and for your careful reading of our manuscript. We have added both references to the sentence that describes gentamicin-induced calcium dysregulation in Results Section 2.4 and have inserted a brief note on connexin 26.
Page 5, lines 200-202 Olfactory receptors, which belong to a family of G protein-coupled receptors (GPCRs) that mediate cellular responses through the activation of activating cAMP signaling, leading to calcium influx [32,33]. This observation agrees with prior research that aminoglycosides disturb Ca²⁺ flux—partly by blocking connexin-26 hemichannels in cochlear cells [34,35]. |
|
Comment 5: Please consider discussion about species specific any polymorphism of TMC1 and OCT2 transporters and how they may alter ototoxicity.
|
Thank you for this valuable comment. We had not considered this aspect in our original discussion, and we greatly appreciate your suggestion. Accordingly, we have added a concise paragraph to the Discussion summarizing the key TMC1 and OCT2 polymorphisms and their potential impact on ototoxicity.
Page 9, lines 357-366 3.5. Species-specific genetic variation Missense mutations in TMC1, such as p.M418K and p.D572N in humans [51,52]and the “Beethoven” p.M412K mutation in mice, can increase MET-channel permeability and heighten sensitivity to aminoglycosides [53]. For OCT2, the common human A270S (rs316019) variant lowers transporter activity and alters cisplatin-related toxicity, whereas wild-type mouse Oct2 has higher drug affinity [23]. Because HEI-OC-1 and UB/OC-2 come from C57BL/6J mice with wild-type alleles, our models reflect baseline uptake rather than variant-specific behavior. Future work with CRISPR-engineered variants or patient-derived iPSC hair cells should clarify genotype-dependent transport and aid personalized otoprotection. |
|
Comment 6: The study had used murine cell lines, is there any particular reason that a rat derived cell line could'nt be used? As most translational work would prefer rat over mouse data especially for drug discovery programs screening for any potential ototoxicity risk for translation to humans.
|
Thank you for this helpful suggestion. We recognize the translational value of rat‐derived models and will consider them in future experiments. At present, however, only a few rat inner-ear lines have been described and they are not yet widely used for ototoxicity screening:
1. Mocha cells—a rat cochlear microglial line immortalized with the E1A gene—were reported in 2017 (Mol. Cell Neurosci. 2017, 86: 58-70) but have seen limited adoption in ototoxicity work. 2. In contrast, three mouse lines—HEI-OC1, UB/OC-1, and UB/OC-2—immortalized with the temperature-sensitive SV40 large-T antigen are well established for aminoglycoside studies (e.g., Hear. Res. 2016, 339: 153-165; Front. Neur. 2021, 12: 725566). These lines benefit from (i) mature mouse transgenic techniques, (ii) controllable differentiation via temperature shift, and (iii) a substantial literature base (>400 publications), which together facilitate reproducibility and accessibility. Rat epithelial or hair-cell–like lines remain scarce and are not yet widely used.
For these practical reasons, mouse auditory cell lines are currently the standard for in-vitro ototoxicity research. Nevertheless, we agree that incorporating rat systems would strengthen translational relevance. We have added a brief statement in the Discussion.
Page 9-10, lines 378-382 We also agree that incorporating rat systems would strengthen translational relevance; therefore, integration with ex vivo rat cochlear explants and in vivo rodent models, followed by functional hearing assessments (e.g., ABR, DPOAE), will be critical for confirming the efficacy of emerging otoprotective strategies. |

Round 2
Reviewer 2 Report
Comments and Suggestions for Authors
The authors have made necessary updates to the manuscript as requested by the reviewer.
For point #3 on using OCT2 inhibitors to validate the role of OCT2 in ototoxicity- the authors plan to do a follow-up work on both MET and OCT2 transporters. Hence, the study with OCT2 inhibitors is not required for this manuscript. Congratulation to all the authors in the manuscript.